

# Intrauterine growth and the maturation process of adrenal function

Sachiko Iwata[1], Masahiro Kinoshita[2], Hisayoshi Okamura[2], Kennosuke Tsuda[1], Mamoru Saikusa[2], Eimei Harada[2], Shinji Saitoh[1] and Osuke Iwata[1]

[1] Center for Human Development and Family Science, Department of Pediatrics and Neonatology, Nagoya City University Graduate School of Medical Sciences, Nagoya, Aichi, Japan
[2] Centre for Developmental and Cognitive Neuroscience, Department of Paediatrics and Child Health, Kurume University School of Medicine, Kurume, Fukuoka, Japan

## ABSTRACT

**Backgrounds:** Environmental factors during early life alter the hypothalamus-pituitary-adrenal (HPA) axis regulation and increase the risk of diseases in later life. However, adrenal function at each developmental stage has not fully been investigated in relation to pathological antenatal conditions. Cortisol levels of newborns with intrauterine growth restriction (IUGR) are elevated during the neonatal period; however, when studied during early childhood, cortisol levels are reduced compared with their peers, suggesting that the HPA axis regulation might be altered from activation to suppression, the timing of which remains uncertain.
**Aim:** The aim of this study was to assess the presence of an interaction between intrauterine growth and postnatal age on cortisol levels in newborns hospitalised at a neonatal intensive care unit.
**Methods:** We performed a secondary analysis using a dataset from saliva samples of 62 newborns collected between 30 and 40 weeks corrected age. Interactions between postnatal age and clinical variables with regard to cortisol levels were assessed.
**Results:** The $z$-score of the birth weight and IUGR showed significant interactions with postnatal age on cortisol levels; cortisol levels were higher $\leq 5$ days of birth and lower $>14$ days of birth than those in their peers without IUGR.
**Conclusion:** The adrenal function of newborns with IUGR might be altered from activation to suppression within the first several weeks of life. Longitudinal studies need to address when/how IUGR alters adrenal functions, and how these responses are associated with diseases during adulthood.

## INTRODUCTION

During pregnancy, plasma cortisol levels markedly change with gestational stage (*Mastorakos & Ilias, 2003*). However, the influence of maternal glucocorticoid on the foetus is limited because of the activation of placental 11-β-hydroxysteroid dehydrogenase 2 (11-β-HSD2) and subsequent conversion of cortisol to biologically inactive cortisone (*Benediktsson et al., 1997*). Clinical conditions such as maternal infection, strict diet,

Corresponding author
Osuke Iwata,
o.iwata@med.nagoya-cu.ac.jp

hypoxia, and foetal stress attenuate placental 11-β-HSD2 activity (*Hardy & Yang, 2002*), which may increase foetal cortisol levels. Foetal exposure to excessive cortisol might induce pathological conditions such as intrauterine growth restriction (IUGR) and preterm labour (*Kajantie et al., 2003*; *Stewart, Rogerson & Mason, 1995*), which are associated with increased risks of short-term mortality and a range of long-term morbidities (*Hack et al., 2002*; *Hutchinson et al., 2013*; *Murray et al., 2015*; *Pallotto & Kilbride, 2006*). Increasing evidence suggests that homeostatic regulation is modified by environmental factors during early life, which is recognised as antenatal programming (*Barker et al., 1990*). Physiological stress response systems, particularly the hypothalamus-pituitary-adrenal (HPA) axis, have been suggested to play a central pathophysiological role in antenatal programming. Although stress-response actions may provide short-term benefits, the resultant alteration of the HPA axis may yield modified responses that result in the development of disease later in life.

Several studies have shown associations between perinatal physiological stress and attenuated adrenal function during childhood. For example, IUGR and preterm birth are associated with attenuated cortisol secretion at ages between 6 months and 7 years (*Brummelte et al., 2015*; *Fattal-Valevski et al., 2005*; *Ruys et al., 2017*). In contrast, when studied during the neonatal period, paradoxically increased cortisol levels were observed in association with IUGR and preterm birth (*Heckmann et al., 1999*; *Kajantie et al., 2003*; *Mericq et al., 2009*; *Stewart, Rogerson & Mason, 1995*), suggesting that the HPA axis regulation might be altered from activation to suppression at some point during infancy or early childhood. However, thus far, few studies have addressed temporal changes in adrenal function following birth in association with exposure to pathological conditions in utero.

Using a cohort of newborns hospitalised in a neonatal intensive care unit (NICU), we have previously reported that lower cortisol levels were associated with clinical events suggestive of unfavourable foetal/postnatal conditions such as exposure to maternal pregnancy-induced hypertension (PIH), caesarean delivery, and preterm birth (*Kinoshita et al., 2016*). Using the same dataset with these studies, we performed a secondary analysis to assess whether maturational changes in cortisol levels after birth vary with the presence of IUGR and other clinical variables suggestive of foetal physiological stress.

## MATERIAL AND METHODS

### Ethics approval and consent

This study was conducted in compliance with the Declaration of Helsinki and with the approval of the Ethics Committee of Kurume University School of Medicine (13142). Written informed consent was obtained from one parent of each participating newborn.

This cross-sectional observational study was conducted as part of a project that investigated the independent variables of cortisol levels and their diurnal patterns (*Kinoshita et al., 2016*, *2018*). In an exploratory analysis of the dataset, the foetal-type diurnal pattern was observed on the basis of salivary cortisol levels, which were entrained in antiphase to the adult rhythm. Furthermore, the analysis found that lower cortisol levels were associated with older postnatal age, maternal PIH, caesarean delivery, preterm

delivery, mechanical ventilation required at birth, and independence from ventilation support on the study day (*Kinoshita et al., 2016*). Feeding-induced elevation of cortisol levels was also noted, which was more prominent following oral feeding and was reduced with prolonged feeding (*Kinoshita et al., 2018*).

## Study population

Detailed data on patient recruitment, sample collection, and cortisol assays have been reported previously (*Kinoshita et al., 2016*, *2018*). In brief, 65 newborns hospitalised in a tertiary NICU were recruited between 30 and 40 weeks (corrected age). Newborns, who were diagnosed with major congenital anomalies or chromosomal aberrations, who had undergone phototherapy within 24 h previously, who had not been weaned from invasive mechanical ventilation and continuous intravenous infusion, and who had received glucocorticoid replacement therapy within 1 week previously, were excluded. Data collection was repeated up to two times for each patient, with a minimum interval of 7 days.

## Sample collection and assay

Saliva was collected before and 1 h after regular feeding at 10:00 and 19:00 using an absorbent swab (SalivaBio; Salimetrics LLC, State College, PA, USA). A cortisol assay was performed using an enzyme immunoassay kit (high-sensitivity salivary cortisol enzyme-linked immunosorbent assay kit; Salimetrics LLC, State College, PA, USA). The limit of detection of this assay in our laboratory was 0.19 nmol/L, and the intra- and inter-assay coefficients of variation were 5.43% and 6.41%, respectively.

## Clinical variables

Clinical information was obtained from electronic patient records, including maternal age, maternal body weight, and body-mass index at the time of delivery, gestational age, delivery mode, intravenous tocolysis, premature rupture of membranes, histopathologically confirmed chorioamnionitis, PIH, multiple gestations, antenatal glucocorticoid administration, birth weight, and its standard score calculated against the standard Japanese birth weight for gestational age (*Itabashi et al., 2014*), IUGR less than the 10th percentile of the norm, sex, Apgar scores, requirement for respiratory support at birth, and predominant type of milk (formula or breast milk) on the day of saliva collection.

## Data analysis

Values are presented as mean (standard deviation), unless otherwise specified. In the current study, only saliva samples collected before (but not after) feeding at 10:00 and 19:00 were considered because of the known influence of feeding on the cortisol levels (*Kinoshita et al., 2018*). Interactions between the clinical variables and postnatal age at saliva collection and their relation to salivary cortisol levels were assessed using the generalised estimating equations with gamma distribution modelling (SPSS ver. 20; IBM, Armonk, NY, USA). To minimise the potential influence of adrenal diurnal patterns and the bias from repeated sampling, the timing of saliva sampling (morning/evening and

**Table 1 Clinical backgrounds of the study cohort.**

| Variables during pregnancy | |
|---|---|
| Maternal and antenatal variables | |
| Intravenous tocolysis | 51 (82.3) |
| Pregnancy-induced hypertension | 10 (16.1) |
| Antenatal glucocorticoid | 29 (46.8) |
| Premature rupture of membranes | 22 (35.5) |
| Chorioamnionitis | 18 (29.0) |
| Maternal age (years) | 31.0 ± 6.2 |
| Maternal height (cm) | 158.2 ± 5.1 |
| Maternal body weight (kg) | 61.1 ± 11.1 |
| Maternal body-mass index | 24.4 ± 4.1 |
| Variables at birth | |
| Vaginal delivery | 26 (41.9) |
| Multiple births | 15 (24.2) |
| Male | 27 (43.5) |
| Gestational age (weeks) | 32.3 ± 3.7 |
| Birth weight (g) | 1671 ± 579 |
| Standard score of the birth weight | −0.55 ± 1.13 |
| Intrauterine growth restriction | 20 (32.3) |
| Apgar score (1 min) | 6.4 ± 2.5 |
| Apgar score (5 min) | 8.1 ± 1.5 |
| Need for mechanical ventilation | 31 (50.0) |
| Postnatal glucocorticoid | 7 (11.3) |
| Variables on the day of saliva sampling | |
| Postnatal age (days) | 24.3 ± 22.6 |
| Corrected age (weeks) | 35.8 ± 1.3 |
| Predominant breast milk feeding | 37 (59.7) |

**Note:**
Values are shown as number (%) or mean ± standard deviation.

the study date when studied on two different days) was incorporated as a within-subject factor. Statistical significance was assumed when $p < 0.003$, after correcting for multiple comparisons of 18 variables. For combinations of categorical variables with significant interactions, the simple effect test was performed for cortisol levels among three postnatal ages of ≤5 days, between 6 and 14 days, and >14 days (defined by the upper and lower quartile) for better clinical translation.

## RESULTS

Out of the 65 newborns, two with an insufficient volume of saliva and one who was subsequently diagnosed with a major chromosomal aberration were excluded from the analysis. Subsequently, 124 samples from 62 cumulative newborns (corrected age, 31.7–39.5 weeks; postnatal age, 1–92 days) who were hospitalised because of low birth weight ($n = 55$), maternal hyperthyroidism or hypothyroidism ($n = 2$), or maternal gestational diabetes mellitus ($n = 5$) were considered (Table 1; Dataset S1).

The main effects of potential antenatal and postnatal physiological stressors on the cortisol levels have been reported previously (See the second paragraph of the Material and Methods section for the brief summary of findings) (*Kinoshita et al., 2016*). Significant interactions were observed between postnatal age and the standard score of the body weight (standardised partial regression coefficient, 1.008; 95% confidence interval, 1.004–1.012; $p < 0.001$), and postnatal age and IUGR (standardised partial regression coefficient, 0.981; 95% confidence interval, 0.968–0.993; $p = 0.002$) (Table 2).

When the simple effect test was performed for newborns with/without IUGR, the cortisol levels were associated with postnatal age in newborns with (but not without) IUGR; cortisol levels at >14 days were lower than those at ≤5 days and between 6 and 14 days (Fig. 1). Subsequently, IUGR newborns showed higher cortisol levels at ≤5 days and paradoxically lower cortisol levels at >14 days than their peers (both $p < 0.001$).

## DISCUSSION

Our study suggested that foetal growth is associated with the relationship between postnatal age and cortisol levels in NICU newborns; cortisol levels of newborns with IUGR were higher soon after birth and lower after two weeks compared to those in their peers without IUGR. These findings build on our previous observations from the same dataset, that maternal PIH, caesarean delivery, and preterm birth are associated with lower cortisol levels in newborns hospitalised in the NICU (*Kinoshita et al., 2016*).

### IUGR and adrenal function after birth

Intrauterine growth restriction is associated with an increased incidence of both short-term and long-term morbidities (*Murray et al., 2015*, *Pallotto & Kilbride, 2006*). In newborns with IUGR, the expression of placental 11-β-HDS2 is decreased (*Stewart, Rogerson & Mason, 1995*), leading to increased cord blood cortisol levels in newborns with low body/placental weight (*Heckmann et al., 1999*; *Mericq et al., 2009*). Increased plasma cortisol levels and angiotensin II receptor subtype 1 expression induce constriction of peripheral vessels, leading to growth restriction of the foetal organs (*Lanz et al., 2003*). Studies further highlighted that antenatal exposure of the foetus to increased cortisol levels contributes to antenatal programming of the HPA axis and other systems, the influence of which is observed throughout (and even after) the developmental process (*Barker et al., 1990*; *Hales et al., 1991*).

Studies involving children and adults found associations between low birth weight and increased basal cortisol levels (*Clark et al., 1996*; *Martinez-Aguayo et al., 2012*). However, when assessed during early childhood, there were conflicting findings, showing no association (*Dahlgren et al., 1998*; *Kajantie et al., 2002*; *Tenhola et al., 2002*) or positive associations (*Elhassan et al., 2015*; *Fattal-Valevski et al., 2005*) between body size at birth and cortisol levels. These inconsistent findings may be derived from the relatively wide age range when cortisol levels were assessed, and the heterogeneous backgrounds of the population, for which the influence of the birth weight was evaluated without incorporating gestational age. Our current data suggest the potential presence of dynamic temporal changes in the adrenal function of IUGR newborns.

**Table 2 Effect of antenatal stressors on cortisol levels according to postnatal age.**

| Clinical conditions | | Cortisol (nmol/L) Mean (SD) | Main effects Condition *p* | PNA *p* | Interaction with PNA β Mean | 95% CI Lower | Upper | *p* |
|---|---|---|---|---|---|---|---|---|
| Intravenous tocolysis | Yes | 8.79 (8.76) | 0.144 | 0.738 | 0.987 | 0.968 | 1.006 | 0.183 |
| | No | 8.03 (5.04) | | | | | | |
| PROM | Yes | 9.24 (9.52) | 0.393 | 0.006 | 0.994 | 0.979 | 1.009 | 0.457 |
| | No | 8.14 (6.51) | | | | | | |
| PIH | Yes | 6.10 (4.21) | 0.259 | *<0.001* | 0.994 | 0.982 | 1.005 | 0.269 |
| | No | 9.00 (8.12) | | | | | | |
| Maternal age | | | 0.442 | 0.598 | 1.000 | 0.999 | 1.001 | 0.962 |
| Maternal body-mass index | | | 0.027 | 0.004 | 1.002 | 1.000 | 1.003 | 0.023 |
| Maternal body weight | | | 0.051 | 0.014 | 1.000 | 1.000 | 1.001 | 0.074 |
| Chorioamnionitis | Yes | 8.41 (6.65) | 0.541 | 0.003 | 1.009 | 0.997 | 1.022 | 0.151 |
| | No | 8.58 (8.11) | | | | | | |
| Antenatal glucocorticoid | Yes | 7.29 (6.15) | 0.819 | 0.008 | 0.993 | 0.979 | 1.008 | 0.367 |
| | No | 9.62 (8,72) | | | | | | |
| Postnatal glucocorticoid | Yes | 6.43 (4.42) | 0.870 | 0.112 | 1.005 | 0.985 | 1.026 | 0.618 |
| | No | 8.80 (7.99) | | | | | | |
| Standard score of the birth weight | | | 0.022 | 0.004 | 1.008 | 1.004 | 1.012 | *<0.001* |
| IUGR | Yes | 9.23 (7.92) | 0.144 | *<0.001* | 0.981 | 0.968 | 0.993 | *0.002* |
| | No | 8.20 (7.60) | | | | | | |
| Multiple pregnancy | Yes | 8.84 (8.74) | 0.756 | 0.182 | 1.004 | 0.982 | 1.026 | 0.744 |
| | No | 8.29 (6.83) | | | | | | |
| Vaginal delivery | Yes | 10.37 (9.52) | 0.641 | 0.013 | 1.009 | 0.997 | 1.020 | 0.140 |
| | No | 7.20 (5.76) | | | | | | |
| Male sex | Yes | 8.29 (6.83) | 0.286 | 0.005 | 0.989 | 0.976 | 1.003 | 0.121 |
| | No | 8.84 (8.74) | | | | | | |
| Gestational age | | | 0.346 | 0.784 | 1.000 | 0.997 | 1.002 | 0.748 |
| 5-min Apgar score <7 | Yes | 6.30 (4.56) | 0.295 | 0.059 | 1.008 | 0.992 | 1.024 | 0.338 |
| | No | 8.86 (8.02) | | | | | | |
| Mechanical ventilation | Yes | 6.76 (5.07) | 0.632 | 0.896 | 0.989 | 0.954 | 1.026 | 0.571 |
| | No | 10.30 (9.34) | | | | | | |
| Breast milk | Yes | 6.88 (5.52) | 0.061 | 0.032 | 1.006 | 0.992 | 1.020 | 0.371 |
| | No | 10.97 (9.64) | | | | | | |

**Notes:**
The *p*-values are from the generalised estimating equation (see "Data analysis" in "Material and Methods" for detail), and are presented for the main effects of each independent variable and postnatal age and their interaction. Statistical significance was assumed when *p* < 0.003, after correcting for multiple comparisons of 18 variables (indicated in bold and italic).

β, standardised partial regression coefficient; IUGR, intrauterine growth restriction; PIH, pregnancy-induced hypertension; PNA, postnatal age; PROM, premature rupture of membranes; SD, standard deviation.

Spontaneous adrenal functioning might be suppressed in growth-restricted foetuses, which causes plasma cortisol levels to diminish rapidly after birth when maternal cortisol is no longer provided.

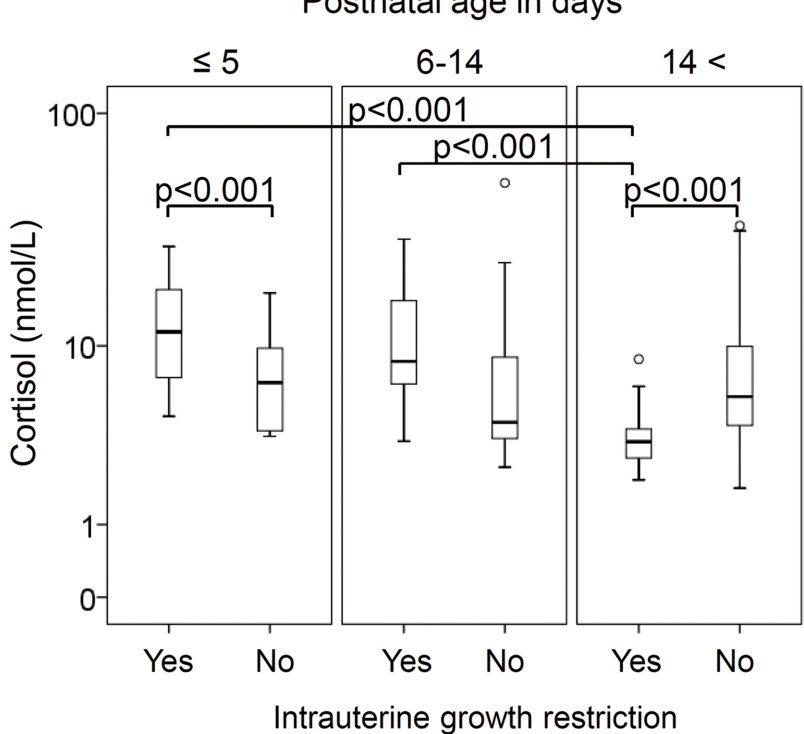

**Figure 1 Box plot depicting the interaction between postnatal age and intrauterine growth restriction on cortisol levels.** IUGR newborns showed higher cortisol levels at ≤5 days and lower cortisol levels at >14 days compared with their peers. Association between postnatal age and cortisol levels was observed for newborns with IUGR, but not for those without IUGR. Statistical findings are from the simple effect test with Bonferroni correction. Symbols: box, first and third quartiles; bold line, median; perpendicular line, range without outliers; and open circle, extreme outlier less than 1.5 times the interquartile range from the first quartile. Abbreviation: IUGR, intrauterine growth restriction.

Because of the observations suggesting accelerated organ maturation of IUGR newborns (*Amiel-Tison et al., 2004*), the rapid postnatal decrease in cortisol levels in IUGR newborns might merely be a hint of the precocious maturation of the HPA axis that occurs with the earlier-than-usual physiological postnatal decrease in cortisol levels. This speculation might be relevant considering previous findings of attenuated HPA functioning during infancy and activated HPA functioning during adolescence and thereafter in IUGR newborns (*Clark et al., 1996*; *Fattal-Valevski et al., 2005*; *Martinez-Aguayo et al., 2012*).

## Limitations of the study

We did not serially obtain multiple saliva samples; temporal changes in cortisol levels were only extrapolated from cross-sectional findings. Because newborns were recruited when their feeding was established, preterm newborns had older postnatal ages than did their near-term/term peers at the time of the study. Our findings need to be re-assessed in future studies, which serially collect saliva samples from the first day of life to at least four weeks. IUGR newborns in the current study cohort exclusively had asymmetrical

IUGR, presumably because symmetrical IUGR was excluded from the study cohort when newborns with major congenital anomalies were excluded.

## CONCLUSIONS

Cortisol levels of newborns were associated with both postnatal age and intrauterine growth. Cortisol levels of newborns with IUGR were higher soon after birth and lower after two weeks than their peers without IUGR. It is possible to speculate that intrauterine growth may alter age-specific cortisol levels via deactivation of the HPA axis within first weeks of life. Future studies should address the impact of IUGR and other perinatal physiological stressors (and their level and duration) on the adrenal function and the development of diseases during adulthood. The potential benefits of care designed to minimise perinatal physiological stressors in these vulnerable newborns also need to be evaluated.

## ACKNOWLEDGEMENTS

The authors thank the patients who participated in the study and their parents for their cooperation, the nurses of the Neonatal Intensive Care Unit, Kurume University Hospital for their support, and Ms Chiho Yoshii and Chiaki Ueno for their consistent support.

### Funding

This work was supported by the Japan Society for the Promotion of Science, The Ministry of Education, Culture, Sports, Science, and Technology (Grant-in-Aid for Scientific Research C16K09005 and C24591533). The funders had no role in study design, data collection and analysis, decision to publish, or preparation of the manuscript.

### Grant Disclosures

The following grant information was disclosed by the authors:
Japan Society for the Promotion of Science, The Ministry of Education, Culture, Sports, Science, and Technology: Grant-in-Aid for Scientific Research C16K09005 and C24591533.

### Competing Interests

The authors declare that they have no competing interests.

### Author Contributions

- Sachiko Iwata conceived and designed the experiments, performed the experiments, analysed the data, contributed reagents/materials/analysis tools, prepared figures and/or tables, authored or reviewed drafts of the paper, approved the final draft.
- Masahiro Kinoshita performed the experiments, analysed the data, contributed reagents/materials/analysis tools, authored or reviewed drafts of the paper, approved the final draft.

- Hisayoshi Okamura performed the experiments, analysed the data, authored or reviewed drafts of the paper, approved the final draft.
- Kennosuke Tsuda performed the experiments, analysed the data, contributed reagents/materials/analysis tools, prepared figures and/or tables, authored or reviewed drafts of the paper, approved the final draft.
- Mamoru Saikusa performed the experiments, prepared figures and/or tables, authored or reviewed drafts of the paper, approved the final draft.
- Eimei Harada performed the experiments, analysed the data, contributed reagents/materials/analysis tools, prepared figures and/or tables, authored or reviewed drafts of the paper, approved the final draft.
- Shinji Saitoh conceived and designed the experiments, authored or reviewed drafts of the paper, approved the final draft.
- Osuke Iwata conceived and designed the experiments, performed the experiments, authored or reviewed drafts of the paper, approved the final draft.

### Human Ethics
The following information was supplied relating to ethical approvals (i.e., approving body and any reference numbers):

This study was conducted in compliance with the Declaration of Helsinki and with the approval of the Ethics Committee of Kurume University School of Medicine (13142).

### Data Availability
Raw data are available as a Supplementary File.

### Supplemental Information
Supplemental information for this article can be found online at http://dx.doi.org/10.7717/peerj.6368#supplemental-information.

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
