# Peer review of "Intrauterine growth and the maturation process of adrenal function"

_PeerJ, doi:10.7717/peerj.6368_

## Round 0.1 · original submission · Major Revisions

Methods and Statistical analysis should be improved, as well as evidence, background and rationale of the study.

·

Basic reporting

In general fine -

a number of small issues around the raw data:

In the raw data there is an unrealistic number of digits in some measurements that go far beyond the actual accuracy of the measurement.
In the raw data the units of the measurements are not given.
In the raw data, the column steroid use is present twice

and two small textual issues:
l.69 What is meant with ‘the remote period’?
l.177: ‘recent studies’ refers to two papers that are >20 years old

Experimental design

Descriptive, cross sections, shortcomings are discussed properly.

Validity of the findings

no comment

Additional comments

This manuscript describes the salivary cortisol levels in newborn infants, as a function of postnatal age (< 5d; > 14 d), and a number of aspects during pregnancy and first days of life, focussing on complications. The main findings are a decrease in cortisol levels over time, and quite strong interactions between age and maternal high blood pressure/ intrauterine growth retardation / C-section. In these three conditions cortisol is higher shortly after birth, and much lower after the age of two weeks.
The authors suggest that these changes indicate reprogramming of the hypothalamus-pituitary-adrenal axis, and that this may be relevant for disease vulnerability in later life.

The manuscript represents a further analysis of earlier reported studies. It is purely descriptive but seems well-controled (this reviewer is not a methodological specialist in this area). And: the research question is in fact a very interesting one, and bears relevance beyond the acute setting of the obstetrics ward. There is a substantial research community focusing on HPA axis changes in relation to adult stress (PTSD) and on ‘early life stress’ in relation to behavioural changes. The initially higher cortisol levels, developing towards lower levels may well bear relevance to those fields.

I have only very minor comments on the manuscript as is.

It struck me that glucocorticoid therapy (outside the exclusion period of one week earlier) has no significant impact by itself. Can the authors discuss this?
It would also be of interest to know what the effects of GCs would be at shorter notice (apparently a week suffices for the HPA axis to normalize, but how fast does this happen?). But probably these children were excluded beforehand, and no measurements are available.

l.69 What is meant with ‘the remote period’?

l.177: ‘recent studies’ refers to two papers that are >20 years old

Reviewer 2 ·

Basic reporting

The article failed to meet in standards in providing enough evidence, rationale and background for both the study rationale and interpreration of the results.

Overall the article flow was hard to follow

Experimental design

The research question needed to be reworded for clarity (i.e. maternal stress, see comments below) and it wasn't clear how it filled a gap nor was substantively different from their previously published work

Methods and Statistical Analysis needed more detail

Validity of the findings

Impact and novelty of data weren't properly discussed and test statistics were missing for all presented data

Additional comments

Title
1) Doesn’t seem IV is antenatal stress, but rather prenatal clinical conditions- see comment 15 as well
2) Association is also likely a more accurate term than Influence.
Abstract
3) In Intro, permanently make be too strong of a word as the HPA system is plastic, perhaps long-term consequences or alters may be more accurate terms
4) In Aims, Would not use the word dependence, but rather association
5) In Methods, would be useful to know when the infant cort levels were collected since results were analyzed overtime and that is a strength of this study
6) The relationship in the Results is not clear
7) The conclusion is concise and clear
Introduction
8) Should be a space between the last word preceding a citation
9) Lines 47 and 48 should be clear if referring to maternal or fetal cortisol/glucocorticoid levels
10) There should be a brief discussion of the consequences of IUGR and preterm labor to strengthen the impact of the current study
11) More rationale is need at the beginning of the second paragraph to understand why looking at the temporal changes in adrenal function are needed (lines 61-63).
12) Add the word exposure in line 63 before pathological and in line 67 before maternal
13) Line 64, it is unclear what authors mean by sporadic observations- perhaps cross-sectional is a better term?
14) Not clear what remote period refers to in line 69.
15) More citations are needed to demonstrate that “several studies have shown associations between perinatal stress and attenuated adrenal function” on lines 68-69. Only one study is cited and more evidence is needed. Also, It is unclear if maternal stress is the correct word to use here (and throughout the manuscript), as the authors are not referring to psychological stress, but rather than physiological stress or maternal health conditions. This should be made clear.
16) The last point on lines 70-74 is not clear. The conclusion of a rapid postnatal decline in cortisol levels does not seem to be supported by the first half of the sentence. Instead, it would be more helpful for the authors to speculate on why this juxtaposition may occur in prenatal vs. postnatal cortisol in offspring with evidence from the literature.
17) Again, the aim that the changes in postnatal cort will change in response to antenatal stressors, does not seem to be the correct term to orient readers the tested research questions.
18) Overall a stronger case needs to be made to look at the interactions with postnatal age and maternal clinical outcomes on infant (see comment 11)
Material and Methods
19) Line 86, the cortisol levels are referred to as independent variables when in fact they are the outcome, thus dependent variables. Also diurnal rhythms aren’t addressed here.
20) It is unclear what is meant by lines 87-91 and how they related to the current study. It is odd to include an exploratory analysis in a Methods section. How did this work inform the current study?
21) Under study population, if the recruitment, sample collection (amount) and cort assay have been reported previously then that needs to be clearly cited. And sample size for this cohort needs to be included here.
22) The times, means ± sd of the ages of saliva collection for the infants should be included in the Methods
23) Was only one saliva sample collected? This seems not in concert with the above statement from the exploratory analysis that suggests a fetal-type diurnal pattern (again unclear what this means and is especially curious because much other work suggests infants don’t develop a diurnal cort pattern right away)
24) Why was postnatal age grouped and not used as a continuous variable?
25) The description of how clinical variables were operationalized needs to be clearer. It seems each clinical variable was tested along with postnatal age on infant cortisol as a multiple comparison correction was used? Is this correct? What stats were used? Was a Two Way ANOVA- clinical dichotomous outcome x postnatal age category -on cortisol levels or change in cortisol levels performed? The Statistical analysis section needs much clarification.
Results
26) Table 1 does not include a full demographic table. Were any demographics considered as covariates like maternal age and/or BMI? Also, it isn’t clear if the numbers refer to percentages or whole numbers and where the comparison numbers are located?
27) The large postnatal range seems to be an issue, especially if postnatal cortisol decreases quickly as suggested and samples were collected within a week apart.
28) The main effects should not be mentioned or reported here as they were previously. These should be included in the introduction and/or discussion where a stronger case needs to be made for this study.
29) Both the tables and text are missing test statistics for the statistical test run (ie, F-value, df, etc).
30) The description of the interactions were unclear, again with the use of the word dependence.
31) Table 2 should report the non-transformed means for cortisol for comparison with other studies, especially since Figure 1 (or Table 3?) has the ln transformed means and comparisons and currently is just a replica of Figure 2 in graph form- thus repeating data.
Discussion
32) Begin the Discussion with a summary of results from the current study, not previous work. Though authors can go on to describe how these fit together. The current summer is still confusing as mentioned in the Abstract presentation of the results above.
33) The importance of IUGR is included here, but would also be useful in the Introduction
34) The second paragraph in the Discussion is nicely written, though overall the Discussion is hard to follow. For example, facts presented in line 173 and 175/176 seem unrelated and their inclusion isn’t clear nor is the relevance to the current study clear.
35) The third paragraph does not include any references nor refer how this work compares with other studies

---

## Round 0.2 · accepted · Accept

The authors have addressed, in details, all the concerns raised by the reviewers.

# ·

Basic reporting

No comments - I found it clear.

Experimental design

I am not a specialist on the type of analysis performed here.

Given the weight that is given in the literature to early life regulation of the HPA axis this is very interesting data in a carefully considered naturalistic setup.

Validity of the findings

The results are discussed in a careful and conservative manner.

I have to refer to other reviewers for judgement of the statistical analysis